# Assessment of availability, awareness and perception of stakeholders regarding preschool vision screening in Kumasi, Ghana: An exploratory study

**Kwadwo Owusu Akuffo**📧\*, **Mohammed Abdul-Kabir, Eldad Agyei-Manu, Josiah Henry Tsiquaye, Christine Karikari Darko, Emmanuel Kofi Addo**

Department of Optometry and Visual Science, College of Science, Kwame Nkrumah University of Science and Technology, Kumasi, Ghana

\* akuffokwadwoowusu@knust.edu.gh, koakuffo@gmail.com

## Abstract

### Background

Regardless of the importance of preschool vision screening (PSVS), there is limited data on the current state of these programs in Africa (particularly Ghana). This study sought to investigate the level of awareness and perception of stakeholders regarding PSVS, its availability and related policies/programmes in the Kumasi Metropolis, Ghana.

### Methods

This descriptive cross-sectional study included 100 systematically sampled preschools in the metropolis (using probability proportional-to-size method); 72 private schools and 28 public schools. Convenience sampling was used to recruit stakeholders of preschools (teachers, head teachers, proprietors, administrators, directors, and educationists), and were interviewed using a well-structured questionnaire. Questionnaires were administered to all eligible respondents who were present at the time of data collection.

### Results

A total of 344 respondents participated in the study; 123 (35.8%) males and 221 (64.2%) females. The overall mean age of respondents was 37.63 ±12.20 years (18–71 years). Of the respondents, 215 (62.5%), 94 (27.3%), and 35 (10.2%) were enrolled from private schools, public schools, and Metropolitan Education Directorate, respectively. 73.8% of respondents reported the absence of routine PSVS in schools whereas 90.1% reported no written policies for PSVS in schools. Only 63.6% of respondents were aware of PSVS whereas more than half (59.6%) of all respondents perceived PSVS to be very important for preschoolers. Private school ownership was significantly associated with availability of PSVS whereas age, teachers, private school ownership, and preschool experience > 10 years were significantly associated with awareness of PSVS ($P$ < 0.05). However, there was no significant association between sociodemographic factors and perception of PSVS.

**Data Availability Statement:** A minimal anonymized data set necessary to replicate study

findings is available in the Supporting Information files.

**Funding:** The author(s) received no specific funding for this work.

**Competing interests:** The authors have declared that no competing interests exist.

**Abbreviations:** CHRPE, Committee on Human Research, Publication & Ethics; HKC, Healthy Kids Check; IAPB, International Agency for the Prevention of Blindness; KNUST, Kwame Nkrumah University of Science and Technology; NECU, National Eye Care Unit; NGO, Non-Governmental Organization; WHO, World Health Organization.

## Conclusion

PSVS is largely unavailable in most Ghanaian schools. Majority of stakeholders were aware of PSVS and agreed to its implementation and incorporation into schools' health programmes. There is the need to implement a national programme/policy on preschool vision screening in Ghana.

## Background

Children across the globe are faced with many visual disorders during the critical periods of their visual development. The most common visual anomalies which affect children, especially preschoolers, include strabismus, amblyopia, large uncorrected refractive errors, and its related risk factors [1, 2]. Globally, the estimated prevalence of strabismus and amblyopia in children is 1.78% and 1.63% respectively [3, 4], whereas the estimated prevalence of hyperopia, myopia, and astigmatism in children is 4.6%, 11.7%, and 14.9%, respectively [5]. Although good vision is necessary for the overall development of these children, these visual anomalies reduce the quality of life of affected children (decreased learning/educational abilities and decreased motor skills performance) [6]. Therefore, early detection and treatment of these visual disorders, through preschool vision screening, is vital in preserving the visual function of children and improving their general health.

The importance of preschool vision screening cannot be underestimated. The introduction of a joint policy statement on vision screening by Practice and Medicine [7] has provided basic recommendations for vision screening in children. Preschool vision screening programmes and policies have been implemented in many countries, including the United States [8], Canada [9], United Kingdom [10], Sweden [11], Australia [12], and South Africa [13]. These vision screenings identify the causes of ocular morbidity in children and offer the best treatment/ management practice, as well as ensuring the consistent availability of vision screening to school-aged children. Childhood vision screening policies and programmes such as the recommendations of the US Preventive Services Task Force [14], the National Children's Vision Screening Project in Australia [12], and the South African Integrated School Health Policy [15] recommends vision screening at least once in all children aged three to five years to detect and manage visual disorders (amblyopia, strabismus, refractive errors, and its risk factors). On the contrary, there are no such nationally adopted policies and programmes on childhood vision screening in Ghana, especially in schools.

The stakeholders involved in preschool vision screenings include parents, school teachers and other educationists, and health workers (such as optometrists, ophthalmologists, etc.). The level of awareness, perception, and responsibilities of these stakeholders plays a vital role in the effectiveness of preschool vision screening and the development of its policies/programmes in schools [16–18]. A study by Senthilkumar, Balasubramaniam [19] showed that although most parents were aware of childhood visual disorders, these parents were unaware of amblyopia in their children, and did not understand the causative factors of many pediatric visual anomalies. In a study by Su, Marvin [20], 29% of parents were unaware of their children's vision screening failure, thereby serving as a barrier to follow-up eye care for their children. Parents perceive that there is inadequate vision screening programmes for their children in various schools [21]. It has also been shown that some teachers are unaware of childhood visual anomalies [22]. This may be due to a lack of education on children's eye health. A study by Agrawal, Tyagi [23] also reported that 96% of teachers were unaware of the age at which vision

screening should be conducted for children. On the other hand, it is interesting to note that some parents and teachers have been educated and trained to improve their awareness on the importance of vision screening [24].

In Ghana, however, stakeholders' perception and awareness on preschool vision screening, as well as the availability of preschool vision screening and its policies/programmes have not been studied extensively. Though various studies have estimated the prevalence of visual disorders among school children and have highlighted the need for measures to address this public health concern [25–27], these studies did not report on the level of awareness and perception of various stakeholders regarding preschool vision screening and its availability. Availability and awareness of preschool screening programmes in any country is the first step to the identification and management of visual conditions among preschoolers, and the consultative process necessary for the formulation of eye health policies for public health impact. The objective of this study is to investigate the level of awareness and perception of stakeholders regarding preschool vision screening and its availability in schools in the Kumasi Metropolis, Ghana. In addition, the outcome of this study will provide recommendations that will inform policy implementation regarding preschool vision screening in Ghanaian schools.

## Materials and methods

This study employed a descriptive cross-sectional design to assess the level of awareness and perception of stakeholders regarding preschool vision screening in some selected preschools. From a list of all schools in the metropolis (obtained from the Metropolitan Education Directorate), schools were classified by the type of school ownership; private schools and public schools. Out of the 412 private and 158 public schools in the metropolis, 100 schools were systematically sampled for this study using Probability Proportional-to-Size (PPS) method; 72 private schools and 28 public schools. This provided a true representation of all preschools within the metropolis. School teachers, head teachers, proprietors, administrators, and directors from these selected preschools within the Kumasi metropolis, as well as educationists from the Metropolitan Education Directorate, Kumasi metropolis, Ghana, participated in this study. All respondents enrolled at the time of this study were eligible to participate in key informant interviews. Respondents from schools and the education directorate were selected using convenience sampling (due to their extracurricular activities). Thus, respondents who were absent at the time of data collection were excluded from this study. The sample size was calculated using the following assumptions/formula: $n = \frac{z^2(p)(1-p)}{d^2}$ (where n = sample size, Z = the standard score at 1.96 for a 95% confidence interval, p = the anticipated prevalence of perceptions of teachers and nurses [estimated to be 0.4 from the study conducted by (Naidoo et al., 2017)], d = absolute error taken as 5%), a minimum sample of 368.79 was estimated. Thus, the target sample size was 370 respondents. Fig 1 shows the flow diagram for the selection of participants through the research process.

### Data collection

The data was obtained from all participants using a well-structured interviewer-administered questionnaire, containing both closed and open-ended questions. The questionnaires were administered by one (1) Principal Investigator and four (4) Research Assistants in English. In each school, questionnaires were administered to the Head of School (e.g. head teacher/proprietor/director/administrator) and/or at least two other teachers in charge of preschools. Questionnaires were also administered to all officials of the Metropolitan Education Directorate who were present at the time of data collection. Each respondent answered each question appropriately after key terms were explained and instructions given.

## Statistical analysis

Statistical analysis was performed using Statistical Product and Service Solution (IBM Corporation IBM® SPSS® Statistics for Windows, Version 23.0 Armonk, NY) compatible with Windows 10. Descriptive statistics were used to determine frequencies and percentages of

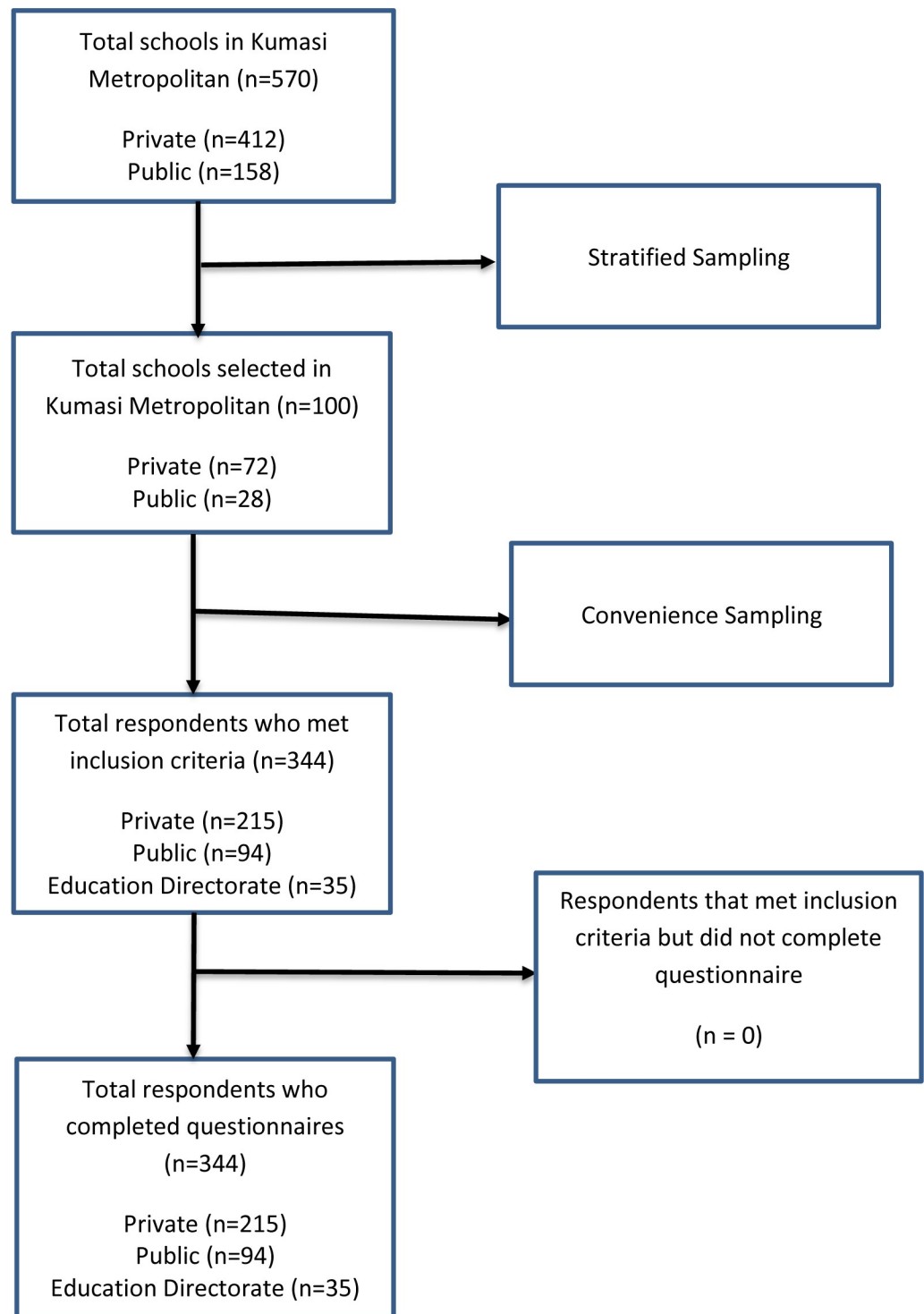

**Fig 1. Flowchart showing the flow of participants through the research process.**

demographic characteristics, awareness and perception of all respondents, as well as the availability of preschool vision screening. Assessment for the perception of preschool vision screening was measured using a five-point Likert scale, which elicited needed responses from participants in the study. Logistic regression analysis further investigated the association between sociodemographic characteristics of respondents and availability, awareness and perception of preschool vision screening. Statistical significance was set at $P < 0.05$.

### Ethical approval

The study was conducted with adherence to the Declaration of Helsinki, and approval was sought from the Committee on Human Research, Publication & Ethics (CHRPE), Kwame Nkrumah University of Science and Technology (KNUST), Kumasi, Ghana (CHRPE/AP/222/18). Permission was also obtained from the Metropolitan Education Directorate and selected schools in the Kumasi Metropolis, Ghana. Verbal informed consent was obtained from all respondents after the details of the nature of the study were explained to them. Verbal consent was deemed given upon acceptance to fill the study questionnaire after receiving information (objectives, procedures, benefits, etc.) on the study, and subsequent completion of the data collection form willingly. Verbal consent was witnessed by the principal investigator (J.H.T.).

## Results

In all, 344 respondents participated in the study; 123 (35.8%) males and 221 (64.2%) females. The overall mean age of all respondents was 37.63 ± 12.20 years (18–71 years), with 215 (62.5%), 94 (27.3%), and 35 (10.2%) being enrolled from private schools, public schools, and Metropolitan Education Directorate, respectively. Most of the respondents (66.3%) had been educated to the tertiary level, with the highest representation being teachers (63.1%), followed by head teachers (21.8%), education directorate respondents (10.2%), directors (2.0%), administrators (1.5%) and proprietors (1.5%). Table 1 reports the demographic characteristics of respondents enrolled in the study.

In the assessment of the availability of preschool vision screening programmes and/or policies, majority of the respondents (73.8%) reported that routine preschool vision screening programmes were not organized in preschools. Although 25.3% of the total respondents reported that disability assessment formed part of the admission processes in schools, 69.4% of these respondents further reported that vision assessment was excluded from the disability assessment programmes in these schools. 310 out of the 344 respondents (90.1%) reported that no written policies for preschool vision screening are available in schools. Out of the schools with written policies for preschool vision screening, 41.0% of respondents reported that these policies have been in existence for a period of 3 to 4 years. In the logistic regression, private school ownership (Odds Ratio [2.46]; 95% CI [1.34–4.51]) was significantly associated with the availability of preschool vision screening (see Table 2). Thus, respondents from private schools were more likely to report the availability of preschool vision screening in their schools.

The level of awareness of respondents regarding preschool vision screening was examined in the study. Of the 344 respondents enrolled in the study, 63.6% reported to have heard about preschool vision screening from eye care professionals (particularly optometrists). Of these, most of the respondents (59.0%) were females. The logistic regression analysis further showed that age (1.03; 1.01–1.05), private school ownership (1.98; 1.14–3.43), teachers (0.46; 0.22–0.97) and preschool experience of more than 10 years (3.45; 1.35–8.85) were statistically significantly related with the awareness of preschool vision screening (see Table 2). Thus, respondents from private schools, teachers, age, and respondents having more than 10 years of preschool experience had an increased likelihood of being aware of preschool vision screening

**Table 1. Demographic characteristics of respondents.**

| Variable | n (%) |
|---|---|
| **Sex** | |
| Male | 123 (35.8) |
| Female | 221 (64.2) |
| *****Age (years)** | |
| ≤ 18 | 4 (1.2) |
| 19–29 | 94 (28.1) |
| 30–39 | 88 (26.3) |
| 40–50 | 84 (25.1) |
| 51–60 | 58 (17.3) |
| ≥ 61 | 7 (2.0) |
| *****Type of School Ownership** | |
| Public | 94 (27.3) |
| Private | 215 (62.5) |
| **Job Title** | |
| Head Teachers | 75 (21.8) |
| Teachers | 217 (63.1) |
| Proprietors | 5 (1.5) |
| Directors | 7 (2.0) |
| Administrators | 5 (1.5) |
| Education directorate respondent | 35 (10.2) |
| **Highest Level of Education** | |
| Primary | 9 (2.6) |
| Secondary | 107 (31.1) |
| Tertiary | 228 (66.3) |
| *****Pre-school Teaching Experience (years)** | |
| 0–1 | 37 (12.0) |
| 2–5 | 116 (37.5) |
| 6–10 | 82 (26.5) |
| More than 10 | 74 (23.9) |
| **Highest Level of School** | |
| Preschool | 16 (4.7) |
| Primary | 162 (47.1) |
| Junior High | 122 (35.5) |
| Senior High | 7 (2.0) |
| Others | 37 (10.8) |
| *****School Enrolment** | |
| Less than 100 | 18 (5.9) |
| 101–200 | 42 (13.7) |
| 201–500 | 140 (45.8) |
| 501–1000 | 84 (27.5) |
| More than 1000 | 22 (7.2) |

n (%) represents the frequencies and percentages of demographic characteristics of respondents.

*n $\neq$ 344

($P < 0.05$, for all). Table 2 reports the association between demographic characteristics of respondents and the availability/awareness/perception of preschool vision screening in the Kumasi Metropolis.

**Table 2. Association between sociodemographic factors and availability/awareness/perception of preschool vision screening.**

| Variable | Availability | | | Awareness | | | Perception | | |
|---|---|---|---|---|---|---|---|---|---|
| | OR | 95% CI | *P*-value | OR | 95% CI | *P*-value | OR | 95% CI | *P*-value |
| **Age (years)** | 1.00 | 0.98–1.02 | 0.763 | 1.03 | 1.01–1.05 | 0.002 | 1.09 | 0.93–1.28 | 0.304 |
| **Sex** | | | | | | | | | |
| Males | ref | - | - | ref | - | - | ref | - | - |
| Females | 1.03 | 0.63–1.67 | 0.921 | 0.71 | 0.45–1.12 | 0.144 | - | - | 0.997 |
| **Education** | | | | | | | | | |
| Primary | ref | - | - | ref | - | - | ref | - | - |
| Secondary | 0.61 | 0.14–2.61 | 0.506 | - | - | 0.999 | - | - | 0.999 |
| Tertiary | 0.90 | 0.22–3.72 | 0.889 | - | - | 0.999 | - | - | 1.000 |
| **School Ownership** | | | | | | | | | |
| Public | ref | - | - | ref | - | - | ref | - | - |
| Private | 2.46 | 1.34–4.51 | 0.004 | 1.98 | 1.14–3.43 | 0.015 | 2.17 | 0.13–35.17 | 0.584 |
| **Job title** | | | | | | | | | |
| Education directorate respondent | ref | - | - | ref | - | - | ref | - | - |
| Headteacher | 0.97 | 0.41–2.30 | 0.936 | 1.81 | 0.80–4.09 | 0.154 | - | - | 1.000 |
| Teacher | 0.76 | 0.35–1.65 | 0.486 | 0.46 | 0.22–0.97 | 0.042 | - | - | 0.998 |
| Proprietor | 1.46 | 0.21–9.98 | 0.703 | 2.25 | 0.33–15.24 | 0.406 | - | - | 1.000 |
| Director | 2.91 | 0.55–15.28 | 0.207 | 3.75 | 0.64–22.10 | 0.144 | - | - | 1.000 |
| Administrator | 3.27 | 0.48–22.46 | 0.228 | 2.25 | 0.33–15.24 | 0.406 | - | - | 1.000 |
| **School Enrolment** | | | | | | | | | |
| Less than 100 | ref | - | - | ref | - | - | ref | - | - |
| 101–200 | 1.17 | 0.34–3.95 | 0.806 | 1.73 | 0.56–5.32 | 0.340 | - | - | 0.999 |
| 201–500 | 1.19 | 0.40–3.55 | 0.753 | 0.70 | 0.25–1.92 | 0.484 | - | - | 0.999 |
| 501–1000 | 0.81 | 0.26–2.56 | 0.723 | 0.74 | 0.26–2.13 | 0.582 | - | - | 1.000 |
| More than 1000 | 0.59 | 0.13–2.58 | 0.472 | 0.35 | 0.83–1.47 | 0.152 | - | - | 1.000 |
| **Pre-school Experience (years)** | | | | | | | | | |
| 0–1 | ref | - | - | ref | - | - | ref | - | - |
| 2–5 | 1.03 | 0.45–2.36 | 0.947 | 1.85 | 0.74–4.62 | 0.186 | - | - | 0.998 |
| 6–10 | 1.18 | 0.50–2.81 | 0.702 | 2.22 | 0.87–5.69 | 0.096 | - | - | 1.000 |
| More than 10 | 1.07 | 0.44–2.59 | 0.881 | 3.45 | 1.35–8.85 | 0.010 | - | - | 1.000 |

CI, Confidence Intervals; ref, reference group in logistic regression analysis.

In the assessment of the perception of respondents regarding preschool vision screening, more than half of the respondents (59.6%) perceived preschool vision screening to be very important for preschoolers (school children). A greater part of the respondents (60.2%) strongly agreed that preschool vision screening should be implemented in preschools. It is interesting to note that 91.9% of all respondents were willing to consider preschool vision screening as a mandatory aspect of admission processes in schools; although 56.6% of these respondents were extremely ready to help in sustaining the preschool vision screening programme in schools. In the logistic regression analysis, there was no statistically significant association between sociodemographic characteristics of respondents and their perception of preschool vision screening (see Table 2).

## Discussion

This study reports the level of awareness and perception of stakeholders regarding preschool vision screening, its availability and related policies/programmes in Ghanaian schools.

Findings from this study suggest that preschool vision screening programmes and policies are largely unavailable in many Ghanaian schools. Majority of the respondents were aware of preschool vision screenings. Most respondents perceived that preschool vision screening is very important and should be implemented in preschools.

Nearly three-quarters of respondents reported that preschool vision screening is an uncommon practice in Ghanaian schools. Respondents were of the view that preschool vision screening has not been incorporated into the schools' curriculum, and that no written policy exists for Ghanaian preschoolers. Respondents' views were contrary to the government-supported vision screening programmes and policies for children of school-going age in other countries [9, 10, 12, 13]. An important and novel finding in this current study was the observation that children are not screened at the time of preschool enrolment. However, the International Agency for the Prevention of Blindness (IAPB) recommends that school-aged children should be screened every 1–2 years in their various schools, and assessed for reduced visual acuity (using a vision screener) and strabismus [28]. Additionally, the United States Preventive Services Task Force Report (2017) recommends that preschool vision screening should be available at least once in children between 3 and 5 years of age [14], in order to detect and manage early visual disorders in preschoolers (to avoid severe visual impairment in their lifetime). In Australia, the introduction of the Healthy Kids Check (HKC), a government-sponsored health screening programme (including vision assessment) for children aged 4 years, has contributed to the overall eye health of these preschoolers [12]. The National Expert Panel to the National Center for Children's Vision and Eye Health (US) recommends best vision screening procedures (monocular visual acuity testing and instrument-based testing) annually for children aged 3 to 6 years, as well as referrals for complete eye examination by optometrists or ophthalmologists [16]. Although these government-sponsored health screening programmes have proven worthwhile, such programmes and policies on preschool vision screening are absent in Ghana. The unavailability of preschool vision screening programmes/policies in Ghana may be due to barriers such as financial constraints, prolonged duration for screening, uncooperative children, and inadequate eye care providers [29]. Although school vision screenings may be conducted by some eye care professionals in an ad hoc manner, there is still the need for a national programme/policy on preschool vision screening in Ghana. The views of stakeholders (such as those interviewed in the current study) will contribute immensely to the promotion and implementation of vision screening programmes/policies in Ghana.

Majority of the respondents were aware of preschool vision screenings and identified eye care professionals as their main source of information. Thus, eye care professionals (optometrists, ophthalmologists, ophthalmic nurses and opticians) in Ghana have a significant role in childhood vision screening [30], and should therefore be at the forefront of awareness campaigns. Eye care professionals must act as channels of communication, and educate stakeholders on childhood vision screening, childhood eye-related diseases/disorders, and how best they can be managed. Public awareness on preschool vision screening may be achieved through radio broadcast, seminars/conferences, and community-based eye-health outreaches. Eye care professionals must also promote the implementation of school-based vision screening programmes, and advocate for the implementation of a national policy for childhood vision screening. Educating teachers, head teachers, administrators/directors, and proprietors of the various preschools on the importance of preschool vision screening programmes will also go a long way to provide awareness on vision screening programmes in Ghana. This was proven by the statistically significant association between teachers and preschool teaching experience of more than 10 years, and the level of awareness of preschool vision screening. The World Health Organization's Health Promoting Schools structure has been reported to be efficient in enhancing the health status of school children [31]. Teachers play a vital role in the

development of comprehensive school health programmes for children in schools (due to their tertiary education and teaching experiences over the years). Thus, educating and enhancing the professional training/development of teachers and other stakeholders in the area of eye health [32] could promote preschool vision screenings and increase follow-up rates for comprehensive eye examinations among preschoolers [33], thereby creating significant awareness on the need for preschool vision screening programmes. In some countries, teachers are trained as first-line 'eye care providers' to effectively assist in the detection of visual anomalies among children [34–36], although some of these training programmes have recorded lower success rates [37]. Screening by teachers has been reported to be cost-effective [38], especially when there is a greater number of preschoolers. It thus enables students to achieve academic and health goals.

Most of the teachers and other educationists in this study perceived preschool vision screening and its related programmes/policies to be very important for preschoolers. The perception of these respondents is in line with some studies [11, 39, 40] which highlight the importance of childhood vision screening (thus reporting a significant reduction in the prevalence of childhood vision anomalies, particularly amblyopia and its risk factors). The implementation of childhood vision screening programmes/policies in some countries has lessened the effects of childhood vision anomalies [8–10, 12, 14]. This view is shared by most respondents in this study. In Canada, some stakeholders perceive that the introduction of public health nurses within schools could facilitate the implementation of school-based vision screenings [41]. It is therefore not surprising that a greater number of respondents in this study were willing to consider preschool vision screening as a mandatory aspect of admission processes in schools. The introduction of the "Health for All Children" in the United Kingdom, which includes school-based vision screenings as part of primary schools' admission processes, has been an effective vision screening system [42]. Therefore, school-entry vision screening in Ghana will contribute immensely to the overall eye health of preschoolers.

The strength of this study lies in the use of an open-ended questionnaire which elicited more details and/or insight from respondents through an infinite number of likely answers. Limitations in this study include a small sample of stakeholders (mainly teachers and educationists) used in this exploratory analysis. Larger sample size should be considered in future researches to enhance the applicability and generalization of results to other locations and countries. Again, the results may be biased and/or skewed towards a section of stakeholders (teachers and educationists only) associated with preschool vision screening in Ghana and may not represent the views/position of other stakeholders such as parents and eye care professionals. However, this study provides a useful background to stakeholders' views on preschool vision screening in the country.

## Conclusions

This study demonstrates the absence of preschool vision screening and its related programmes/policies in Ghanaian schools. Stakeholders were aware of preschool vision screenings and agreed with the implementation of school-entry vision screenings programmes/policies in Ghanaian schools. Further studies must be conducted to ascertain the level of awareness and perception of parents and eye care professionals regarding preschool vision screening, its availability and related policies/programmes in Ghana. Also, this study will help in developing a suitable preschool vision screening protocol specific to the Ghanaian setting.

This study recommends a broader stakeholder consultation (involving parents, teachers, eye care professionals, heads of eye care institutions, NGO's, etc.) by the National Eye Care Unit (NECU) of the Ghana Health Service regarding school vision screening in Ghana. The

National Eye Care Unit, Ghana Health Service must then develop a proposal on school vision screenings in Ghana based on recommendations from the IAPB and WHO, review of the national school vision screening programmes/policies of other countries, and the views of all stakeholders of school vision screening in Ghana. Subsequently, the proposal for school-based vision screening should be tendered into the Ministry of Health, Ghana for its consideration and implementation.

## Supporting information

**S1 File.**
(SAV)

## Author Contributions

**Conceptualization:** Kwadwo Owusu Akuffo, Mohammed Abdul-Kabir.

**Data curation:** Kwadwo Owusu Akuffo, Josiah Henry Tsiquaye, Christine Karikari Darko.

**Formal analysis:** Kwadwo Owusu Akuffo, Josiah Henry Tsiquaye.

**Methodology:** Kwadwo Owusu Akuffo, Josiah Henry Tsiquaye, Emmanuel Kofi Addo.

**Project administration:** Kwadwo Owusu Akuffo, Mohammed Abdul-Kabir, Eldad Agyei-Manu, Josiah Henry Tsiquaye, Christine Karikari Darko, Emmanuel Kofi Addo.

**Resources:** Kwadwo Owusu Akuffo, Mohammed Abdul-Kabir, Eldad Agyei-Manu, Josiah Henry Tsiquaye, Christine Karikari Darko, Emmanuel Kofi Addo.

**Supervision:** Eldad Agyei-Manu, Emmanuel Kofi Addo.

**Writing – original draft:** Kwadwo Owusu Akuffo, Eldad Agyei-Manu, Josiah Henry Tsiquaye, Christine Karikari Darko, Emmanuel Kofi Addo.

**Writing – review & editing:** Kwadwo Owusu Akuffo, Mohammed Abdul-Kabir, Eldad Agyei-Manu, Josiah Henry Tsiquaye, Christine Karikari Darko, Emmanuel Kofi Addo.

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
