## [Decision Letter · Decision Letter 0]

4 Oct 2019

PONE-D-19-23753

Assessment of Availability, Awareness and Perception of Stakeholders regarding Preschool Vision Screening in Kumasi, Ghana: An Exploratory Study

PLOS ONE

Dear Dr Akuffo,

Thank you for submitting your manuscript to PLOS ONE. After careful consideration, we feel that it has merit but does not fully meet PLOS ONE’s publication criteria as it currently stands. Therefore, we invite you to submit a revised version of the manuscript that addresses the points raised during the review process.

ACADEMIC EDITOR: 

While the manuscript carries some useful information, it needs extensive revision and further details of the methods section as well as the health care system in Ghana

We would appreciate receiving your revised manuscript by Nov 18 2019 11:59PM. To enhance the reproducibility of your results, we recommend that if applicable you deposit your laboratory protocols in protocols.io, where a protocol can be assigned its own identifier (DOI) such that it can be cited independently in the future. For instructions see: http://journals.plos.org/plosone/s/submission-guidelines#loc-laboratory-protocols

We look forward to receiving your revised manuscript.

Kind regards,

Ahmed Awadein, MD, Ph.D, FRCS

Academic Editor

PLOS ONE

Journal Requirements:

2. Please provide additional details regarding participant consent. In the ethics statement in the Methods and online submission information, please ensure that you have specified how verbal consent was documented and witnessed

Additional Editor Comments (if provided):

Reviewers' comments:

Reviewer's Responses to Questions

**Comments to the Author**

1. Is the manuscript technically sound, and do the data support the conclusions?

Reviewer #1: Partly

Reviewer #2: Yes

2. Has the statistical analysis been performed appropriately and rigorously? 

Reviewer #1: No

Reviewer #2: I Don't Know

3. Have the authors made all data underlying the findings in their manuscript fully available?

Reviewer #1: No

Reviewer #2: Yes

4. Is the manuscript presented in an intelligible fashion and written in standard English?

Reviewer #1: Yes

Reviewer #2: Yes

5. Review Comments to the Author

Reviewer #1: This paper is an interesting submission and related to an important area of eye care research on issues around preschool vision screening program. There are many areas that need further clarification, the writing needs to be more succinct. The presentation of the article is somewhat disjointed with some excellent portions and some portions that don't seem to flow as well.

I would like to offer a few suggestions for how the manuscript presentation could be improved:

Introduction:

• The introduction section probably would benefit from some tightening with clear objective of conducting this study and using as recent references as possible.

• The context of the study needs to be explained to situate the findings for international readers to provide a frame of reference for the findings.

• In the discussion, the authors should come back to the literature you cite in the introduction and consider what they need to present in the beginning, to reflect on in the end.

Methods:

• The method section is too brief and the overall research process appears rather complex. The way the authors articulated the method section is a bit unclear. I would suggest making it clear for the readers.

• The process of sample selection - what was the sampling technique? The authors mentioned about systematic sampling (probability proportional-to-size method) to select schools and convenient sampling to select interview participants. However, it is not clear why and how these sampling techniques were used and based on what assumptions.

• How sample size was calculated and numbers mentioned decided based on what assumption/formula? – The authors should mention these clearly in the method section.

• Also the flow of participants through the research process is unclear. It would be good for the readers if the authors specify exactly how the process was done? In order to clarify, could the authors provide a flow diagram (such as recommended by the CONSORT group) so that readers can understand more clearly exactly what the researchers did throughout the research process, sample size, approached/ agreed, included/ excluded participants etc.

• Variable: No information was provided about outcome and explanatory variable. It is also not clear what was the exact outcome measures. Reading through the title and text – it appears that outcome variables were more than one - awareness and perception, and the availability of preschool vision screening. However, none of the variables were defined.

• The analysis section requires more clarity. For example, it is not clear how perception of preschool vision screening was measured. Did researchers use any specific scale for that?

• The authors stated that “Chi-square test was used to determine the association between demographic characteristics of respondents, and their awareness and perception, and the availability of preschool vision screening.” – But no association data is provided for perception, and the availability of preschool vision screening in the results. Moreover, the authors did not run any advance level analysis (bi-variate of multi-variate analysis) to establish any solid association between outcome variables and explanatory variables.

• Ethical issues are now mixed-up with the method section. The authors might consider putting the ethical approval and other ethical issues as a sub-heading at the end of the method section.

Result:

• The findings have not really been fully teased and pondered.

• Whilst I appreciate there is a huge amount of data and much to explore one glaring point of interest to me was further elaboration of results.

• The result section only highlighted the basic findings on demographic characteristic and few information on awareness and perception, and the availability of preschool vision screening. No associated factors are provided for perception, and the availability of preschool vision screening.

• The authors did not run any bi-variate of multi-variate analysis to establish any strong association between outcome variables and explanatory variables. It is difficult to make any inference based on the results provided at present.

• Is it possible to expand the result section further establishing association between outcome variables and explanatory variables through running bi-variate of multi-variate analysis?

Discussion:

• The discussion should be more analytical and should reflect on how the findings support, refute, extend the previous literature (generally cited in the introduction).

• Results of previous studies are simply stated alongside this study with no discussion of why the findings may differ between the studies etc.

• What is implication of the study results for preschool vision screening program and eye care linked clinical practice?

• Moreover, the section would benefit from some tightening through building more precise arguments more in the context of this study findings only.

• The researchers highlight a few limitations of the study which should be further pondered. The strengths of the study should be highlighted as well. The authors should highlight limitations/strengths with strong arguments.

This is an interesting topic and I hope the authors will find these comments useful as they consider how to proceed with the manuscript.

Reviewer #2: I guess the aim of the manuscript is to inform the Ghana Health authority that you are ready to accept any screening policy in case of implementation. In the majority of screening policies implemented throughout the world rest on primary care. Curious to know the health structure in Ghana Do family practitioners perform or would be willing to do the screenings. Any screening program performed in school would be an addition to the primary care physician screening.

In your discussion please state the primary care system in Ghana.

6. PLOS authors have the option to publish the peer review history of their article (what does this mean?). If published, this will include your full peer review and any attached files.

Reviewer #1: Yes: Tapash Roy

Reviewer #2: Yes: Silay Canturk Ugurbas

---

## [Author Response · Author response to Decision Letter 0]

13 Feb 2020

Re: Manuscript PONE-D-19-23753 titled: “Assessment of Availability, Awareness and Perception of Stakeholders regarding Preschool Vision Screening in Kumasi, Ghana: An Exploratory Study”.

Dear Editor, 

Thank you for the opportunity to resubmit the above referenced manuscript. We have addressed all the Reviewers’ comments, point-by-point, in our response below. The issues raised by the Reviewers are presented in normal font. We reply directly to these comments in bold font. All changes to the Revised Manuscript have also been clearly highlighted in red font.

Sincerely, 

Akuffo et al

Journal Requirements:

We have now revised the manuscript according to the PLOS ONE's style requirements.

2. Please provide additional details regarding participant consent. In the ethics statement in the Methods and online submission information, please ensure that you have specified how verbal consent was documented and witnessed

Verbal consent was deemed given upon acceptance to fill the study questionnaire after receiving information (objectives, procedures, benefits, etc.) on the study, and subsequent completion of the data collection form willingly. Verbal consent was witnessed by the principal investigator (J.H.T.). We have included a statement in the Revised Manuscript for clarity. Please see below and Revised Manuscript page 10.

“Verbal consent was deemed given upon acceptance to fill the study questionnaire after receiving information (objectives, procedures, benefits, etc.) on the study, and subsequent completion of the data collection form willingly. Verbal consent was witnessed by the principal investigator (J.H.T.).”

Additional Editor Comments (if provided):

Reviewers' comments:

Reviewer's Responses to Questions

Comments to the Author

1. Is the manuscript technically sound, and do the data support the conclusions?

Reviewer #1: Partly

Reviewer #2: Yes

We have revised the description of our study methodology and have provided further statistical analyses for clarity (Please see below; and Methods Pages 6 to 9; Results 9 to 17 and Discussion Pages 18 to 22).

Methods section:

The expression n= (where n =sample size, Z= the standard score at 1.96 for a 95% confidence interval, p = was the anticipated prevalence of perceptions of teachers and nurses [estimated to be 0.4 from the study conducted by (Naidoo et al., 2017), d= absolute error taken as 5%), a minimum sample of 368.79 was estimated. Thus, the target sample size was 370 respondents. We have added this to the methods section (please see below and Revised Manuscript Pages 6 to 7).

“The sample size was called using the following assumptions/formula: 

The expression n= (where n =sample size, Z= the standard score at 1.96 for a 95% confidence interval, p = was the anticipated prevalence of perceptions of teachers and nurses [estimated to be 0.4 from the study conducted by (Naidoo et al., 2017), d= absolute error taken as 5%), a minimum sample of 368.79 was estimated. Thus, the target sample size was 370 respondents.”

We have now provided a flow diagram detailing the research process. Please see Figure 1 below and in the Revised Manuscript (page 8).

Figure 1. Flowchart showing the flow of participants through the research process

We have revised our Statistical Analysis section and provided additional details. Variables used in the study included socio-demographic data on all participants. These include sex, age, type of school ownership, job title, highest level of education, preschool teaching experience, and school enrolment. 

For the analyses pertaining to awareness of preschool vision screening, the explanatory variables included age, sex, type of school ownership, job title, education, preschool teaching experience. Awareness of preschool vision screening among participants (awareness variable) was assessed by answering the following questions: “Have you heard of Preschool Vision Screening services for preschool children?” The awareness variable was employed in the univariate and multivariate logistic regression models assessing the association between awareness and related factors.

For the analyses pertaining to availability of preschool vision screening, the explanatory variables included age, sex, type of school ownership, job title, education, preschool teaching experience. Availability of preschool vision screening among participants (availability variable) was assessed by answering the following questions: “Does your school(s) have a written policy for screening preschool children? And/or Does your school(s) conduct vision examinations/screenings routinely for preschool children?” The availability variable was employed in the univariate and multivariate logistic regression models assessing the association between awareness and related factors.

For the analyses pertaining the perception, the explanatory variables included age, sex, type of school ownership, job title, education, preschool teaching experience. The perception of preschool vision screening among participants (perception variable) was assessed by answering the following questions: “Do you think that Preschool Vision Screenings should be implemented in all schools? And/or If preschool vision screening is not done in your school(s), do you think it is important for children to benefit from routine eye examinations? And/or Will you consider Preschool Vision Screening mandatory as part of the admission process?” The perception variable is a composite of the above questions. The perception variable was employed in the univariate and multivariate logistic regression models assessing the association between awareness and related factors.

Assessment for the perception of preschool vision screening was measured using the responses of participants in the study. The scale employed was a five-point Likert scale to assess responses of participants. Please see response above under ‘Variable’. We have revised our Statistical Analysis section and provided additional details (please see Revised Manuscript pages 7-8).

Assessment for the perception of preschool vision screening was measured using a five-point Likert scale, which elicited needed responses from participants in the study. 

We have revised our Statistical Analysis section and included a statement in the Revised Manuscript for clarity. Please see below and Revised Manuscript pages 8

“Logistic regression analysis further investigated the association between sociodemographic characteristics of respondents and availability, awareness and perception of preschool vision screening.”

We have now restructured and provided an Ethical Approval Section at the end of the methods (please see below and Revised Manuscript pages 10).

“Ethical Approval

The study was conducted with adherence to the Declaration of Helsinki, and approval was sought from the Committee on Human Research, Publication & Ethics (CHRPE), Kwame Nkrumah University of Science and Technology (KNUST), Kumasi, Ghana (CHRPE/AP/222/18). Permission was also obtained from the Metropolitan Education Directorate and selected schools in the Kumasi Metropolis, Ghana. A verbal informed consent was obtained from all respondents after the details of the nature of the study were explained to them”. 

Results section:

We have provided additional analyses and presented additional results based on our study data (please see Revised Manuscript pages 12 to 17).

“In the logistic regression, private school ownership (Odds Ratio [2.46]; 95% CI [1.34-4.51]) was significantly associated with the availability of preschool vision screening. Thus, respondents from private schools were more likely to report the availability of preschool vision screening in their schools.

The logistic regression analysis showed that age (1.03; 1.01-1.05), private school ownership (1.98; 1.14-3.43), teachers (0.46; 0.22-0.97) and preschool experience of more than 10 years (3.45; 1.35-8.85) were statistically significantly related with the awareness of preschool vision screening. Thus, respondents from private schools, teachers, age, and respondents having more than 10 years of preschool experience had an increased likelihood of being aware of preschool vision screening.

In the logistic regression analysis, there was no statistically significant association between sociodemographic characteristics of respondents and their perception of preschool vision screening.”

We have also provided a new table titled “Table 2. Association between sociodemographic factors and availability/awareness/perception of preschool vision screening”. This new table provides further details on results from logistic regression analyses.

Discussion section:

We have now revised the limitations/strengths section of the Discussion with strong arguments. Please see below and Revised Manuscript Page 21.

“The strength of this study lies in the use of an open-ended questionnaire which elicited more details and/or insight from respondents through an infinite number of likely answers. Limitations in this study include a small sample of stakeholders (mainly teachers and educationists) used in this exploratory analysis. Larger sample size should be considered in future researches to enhance the applicability and generalization of results to other locations and countries. Again, the results may be biased and/or skewed towards a section of stakeholders (teachers and educationists only) associated with preschool vision screening in Ghana and may not represent the views/position of other stakeholders such as parents and eye care professionals.”

2. Has the statistical analysis been performed appropriately and rigorously?

Reviewer #1: No

Reviewer #2: I Don't Know 

We have revised the statistical methodology and provided more results for clarity (Please see below; and Methods Pages 6 to 9; Results 9 to 17).

Statistical Analysis section:

We have revised our Statistical Analysis section and provided additional details. Variables used in the study included socio-demographic data on all participants. These include sex, age, type of school ownership, job title, highest level of education, preschool teaching experience, and school enrolment. 

For the analyses pertaining to awareness of preschool vision screening, the explanatory variables included age, sex, type of school ownership, job title, education, preschool teaching experience. Awareness of preschool vision screening among participants (awareness variable) was assessed by answering the following questions: “Have you heard of Preschool Vision Screening services for preschool children?” The awareness variable was employed in the univariate and multivariate logistic regression models assessing the association between awareness and related factors.

For the analyses pertaining to availability of preschool vision screening, the explanatory variables included age, sex, type of school ownership, job title, education, preschool teaching experience. Availability of preschool vision screening among participants (availability variable) was assessed by answering the following questions: “Does your school(s) have a written policy for screening preschool children? And/or Does your school(s) conduct vision examinations/screenings routinely for preschool children?” The availability variable was employed in the univariate and multivariate logistic regression models assessing the association between awareness and related factors.

For the analyses pertaining the perception, the explanatory variables included age, sex, type of school ownership, job title, education, preschool teaching experience. The perception of preschool vision screening among participants (perception variable) was assessed by answering the following questions: “Do you think that Preschool Vision Screenings should be implemented in all schools? And/or If preschool vision screening is not done in your school(s), do you think it is important for children to benefit from routine eye examinations? And/or Will you consider Preschool Vision Screening mandatory as part of the admission process?” The perception variable is a composite of the above questions. The perception variable was employed in the univariate and multivariate logistic regression models assessing the association between awareness and related factors.

Assessment for the perception of preschool vision screening was measured using the responses of participants in the study. The scale employed was a five-point Likert scale to assess responses of participants. Please see response above under ‘Variable’. We have revised our Statistical Analysis section and provided additional details (please see Revised Manuscript pages 7; lines 130-132).

Assessment for the perception of preschool vision screening was measured using a five-point Likert scale, which elicited needed responses from participants in the study. 

We have revised our Statistical Analysis section and included a statement in the Revised Manuscript for clarity. Please see below and Revised Manuscript pages 7; Lines 132 to 133

“Logistic regression analysis further investigated the association between sociodemographic characteristics of respondents and availability, awareness and perception of preschool vision screening.”

Results section:

We have provided additional analyses and presented additional results based on our study data (please see Revised Manuscript pages 12 to 17).

“In the logistic regression, private school ownership (Odds Ratio [2.46]; 95% CI [1.34-4.51]) was significantly associated with the availability of preschool vision screening. Thus, respondents from private schools were more likely to report the availability of preschool vision screening in their schools.

The logistic regression analysis showed that age (1.03; 1.01-1.05), private school ownership (1.98; 1.14-3.43), teachers (0.46; 0.22-0.97) and preschool experience of more than 10 years (3.45; 1.35-8.85) were statistically significantly related with the awareness of preschool vision screening. Thus, respondents from private schools, teachers, age, and respondents having more than 10 years of preschool experience had an increased likelihood of being aware of preschool vision screening.

In the logistic regression analysis, there was no statistically significant association between sociodemographic characteristics of respondents and their perception of preschool vision screening.”

We have also provided a new table titled “Table 2. Association between sociodemographic factors and availability/awareness/perception of preschool vision screening”. This new table provides further details on results from logistic regression analyses.

3. Have the authors made all data underlying the findings in their manuscript fully available?

The PLOS Data policy requires authors to make all data underlying the findings described in their manuscript fully available without restriction, with rare exception (please refer to the Data Availability Statement in the manuscript PDF file). The data should be provided as part of the manuscript or its supporting information or deposited to a public repository. For example, in addition to summary statistics, the data points behind means, medians and variance measures should be available. If there are restrictions on publicly sharing data—e.g. participant privacy or use of data from a third party—those must be specified.

Reviewer #1: No

Reviewer #2: Yes

We have provided all data underlying our findings in our manuscript. We stated clearly in our submitted manuscript that: “The datasets generated during and/or analyzed during the current study are available from the corresponding author on reasonable request.” We provided this Data Availability statement because we did not include in our ethical approval application that we were going to upload study data on a public repository. 

4. Is the manuscript presented in an intelligible fashion and written in standard English?

Reviewer #1: Yes

Reviewer #2: Yes

We thank the Reviewers for their positive comments on our English grammar.

5. Review Comments to the Author

Reviewer #1: This paper is an interesting submission and related to an important area of eye care research on issues around preschool vision screening program. There are many areas that need further clarification, the writing needs to be more succinct. The presentation of the article is somewhat disjointed with some excellent portions and some portions that don't seem to flow as well.

We thank Reviewer #1 for his/her positive comments on our manuscript. We have addressed the suggestions and questions raised by Reviewer 1 below.

I would like to offer a few suggestions for how the manuscript presentation could be improved:

Introduction:

• The introduction section probably would benefit from some tightening with clear objective of conducting this study and using as recent references as possible.

The objective of our study was “To investigate the level of awareness and perception of stakeholders regarding preschool vision screening, its availability and related policies/programmes in the Kumasi Metropolis, Ghana”. The introduction section clearly highlights gaps in literature available, especially in the Ghanaian context, and hence the objective raised in the study. Most references used in the introduction does not go beyond a decade. We have, however, reviewed our introduction section and revised it accordingly (please see Revised Manuscript pages 4 to 6).

We have also revised our objective for the study to be “The objective of this study is to investigate the level of awareness and perception of stakeholders regarding preschool vision screening and its availability in schools in the Kumasi Metropolis, Ghana”.

• The context of the study needs to be explained to situate the findings for international readers to provide a frame of reference for the findings.

The first paragraph under the Introduction section highlights vision disorders among children globally, especially preschoolers. The second paragraph evaluates the need for early visual assessment for preschoolers and provides examples of national policies by some countries which is aimed at tackling this menace. The third paragraph also highlights the various stakeholders and their possible roles in achieving good vision among preschoolers. The last paragraph reviews these assessments in the Ghanaian context, and the need for further studies and a national policy regarding preschool vision screening in Ghana.

• In the discussion, the authors should come back to the literature you cite in the introduction and consider what they need to present in the beginning, to reflect on in the end.

The literature as used in the Introduction section (references 7, 11-16, 18 from originally submitted manuscript) were also used under the Discussion section to explain findings in our study. We, however, acknowledge other references which were drawn to fully support our arguments under the Discussion section. We have, therefore, reviewed our introduction section and revised it accordingly (please see Revised Manuscript pages 4 to 6).

Methods:

• The method section is too brief, and the overall research process appears rather complex. The way the authors articulated the method section is a bit unclear. I would suggest making it clear for the readers.

The Methods section highlights salient procedures which were employed in our study (considering the word limit requirement by the journal). The procedure for the study were clearly outlined. We however acknowledge Reviewers comment and have provided additional information for clarity. Please see below and Methods Section Pages 6-9; lines 107-168.

The expression n= (where n =sample size, Z= the standard score at 1.96 for a 95% confidence interval, p = was the anticipated prevalence of perceptions of teachers and nurses [estimated to be 0.4 from the study conducted by (Naidoo et al., 2017), d= absolute error taken as 5%), a minimum sample of 368.79 was estimated. Thus, the target sample size was 370 respondents. We have added this to the methods section (please see below and Revised Manuscript Pages 6 to 7; lines 108 - 113)

“The sample size was called using the following assumptions/formula: 

The expression n= (where n =sample size, Z= the standard score at 1.96 for a 95% confidence interval, p = was the anticipated prevalence of perceptions of teachers and nurses [estimated to be 0.4 from the study conducted by (Naidoo et al., 2017), d= absolute error taken as 5%), a minimum sample of 368.79 was estimated. Thus, the target sample size was 370 respondents.”

We have now provided a flow diagram detailing the research process. Please see Figure 1 below and in the Revised Manuscript (page 8).

Figure 1. Flowchart showing the flow of participants through the research process

We have revised our Statistical Analysis section and provided additional details. Variables used in the study included socio-demographic data on all participants. These include sex, age, type of school ownership, job title, highest level of education, preschool teaching experience, and school enrolment. 

Assessment for the perception of preschool vision screening was measured using the responses of participants in the study. The scale employed was a five-point Likert scale to assess responses of participants. Please see response above under ‘Variable’. We have revised our Statistical Analysis section and provided additional details (please see Revised Manuscript pages 7; lines 130-132).

Assessment for the perception of preschool vision screening was measured using a five-point Likert scale, which elicited needed responses from participants in the study. 

We have revised our Statistical Analysis section and included a statement in the Revised Manuscript for clarity. Please see below and Revised Manuscript pages 7; Lines 132 to 133

“Logistic regression analysis further investigated the association between sociodemographic characteristics of respondents and availability, awareness and perception of preschool vision screening.”

We have now restructured and provided an Ethical Approval Section at the end of the methods (please see below and Revised Manuscript pages 9; lines 159-168).

“Ethical Approval

The study was conducted with adherence to the Declaration of Helsinki, and approval was sought from the Committee on Human Research, Publication & Ethics (CHRPE), Kwame Nkrumah University of Science and Technology (KNUST), Kumasi, Ghana (CHRPE/AP/222/18). Permission was also obtained from the Metropolitan Education Directorate and selected schools in the Kumasi Metropolis, Ghana. A verbal informed consent was obtained from all respondents after the details of the nature of the study were explained to them”. 

• The process of sample selection - what was the sampling technique? The authors mentioned about systematic sampling (probability proportional-to-size method) to select schools and convenient sampling to select interview participants. However, it is not clear why and how these sampling techniques were used and based on what assumptions.

Probability Proportional-to-Size sampling technique was employed in the selection of schools available in the Kumasi metropolis. Due to unequal numbers of schools under the various type of school ownership (private schools and public schools), we had to use this method. Convenience sampling was employed in the selection of interview participants because it was assumed that some of the teachers would not be available at the time of data collection (due to their extracurricular activities). The statement “due to their extracurricular activities” has been added under line 107 of the Methods section. 

• How sample size was calculated, and numbers mentioned decided based on what assumption/formula? – The authors should mention these clearly in the method section.

The expression n= (where n =sample size, Z= the standard score at 1.96 for a 95% confidence interval, p = was the anticipated prevalence of perceptions of teachers and nurses [estimated to be 0.4 from the study conducted by (Naidoo et al., 2017), d= absolute error taken as 5%), a minimum sample of 368.79 was estimated. Thus, the target sample size was 370 respondents. We have added this to the methods section (please see below and Revised Manuscript Pages 6 to 7; lines 108 - 113)

“The sample size was called using the following assumptions/formula: 

The expression n= (where n =sample size, Z= the standard score at 1.96 for a 95% confidence interval, p = was the anticipated prevalence of perceptions of teachers and nurses [estimated to be 0.4 from the study conducted by (Naidoo et al., 2017), d= absolute error taken as 5%), a minimum sample of 368.79 was estimated. Thus, the target sample size was 370 respondents.”

• Also, the flow of participants through the research process is unclear. It would be good for the readers if the authors specify exactly how the process was done? In order to clarify, could the authors provide a flow diagram (such as recommended by the CONSORT group) so that readers can understand more clearly exactly what the researchers did throughout the research process, sample size, approached/ agreed, included/ excluded participants etc.

We have now provided a flow diagram detailing the research process. Please see Figure 1 below and in the Revised Manuscript (page 8).

Figure 1. Flowchart showing the flow of participants through the research process

• Variable: No information was provided about outcome and explanatory variable. It is also not clear what was the exact outcome measures. Reading through the title and text – it appears that outcome variables were more than one - awareness and perception, and the availability of preschool vision screening. However, none of the variables were defined. 

We have revised our Statistical Analysis section and provided additional details. Variables used in the study included socio-demographic data on all participants. These include sex, age, type of school ownership, job title, highest level of education, preschool teaching experience, and school enrolment. 

For the analyses pertaining to awareness of preschool vision screening, the explanatory variables included age, sex, type of school ownership, job title, education, preschool teaching experience. Awareness of preschool vision screening among participants (awareness variable) was assessed by answering the following questions: “Have you heard of Preschool Vision Screening services for preschool children?” The awareness variable was employed in the univariate and multivariate logistic regression models assessing the association between awareness and related factors.

For the analyses pertaining to availability of preschool vision screening, the explanatory variables included age, sex, type of school ownership, job title, education, preschool teaching experience. Availability of preschool vision screening among participants (availability variable) was assessed by answering the following questions: “Does your school(s) have a written policy for screening preschool children? And/or Does your school(s) conduct vision examinations/screenings routinely for preschool children?” The availability variable was employed in the univariate and multivariate logistic regression models assessing the association between awareness and related factors.

For the analyses pertaining the perception, the explanatory variables included age, sex, type of school ownership, job title, education, preschool teaching experience. The perception of preschool vision screening among participants (perception variable) was assessed by answering the following questions: “Do you think that Preschool Vision Screenings should be implemented in all schools? And/or If preschool vision screening is not done in your school(s), do you think it is important for children to benefit from routine eye examinations? And/or Will you consider Preschool Vision Screening mandatory as part of the admission process?” The perception variable is a composite of the above questions. The perception variable was employed in the univariate and multivariate logistic regression models assessing the association between awareness and related factors.

• The analysis section requires more clarity. For example, it is not clear how perception of preschool vision screening was measured. Did researchers use any specific scale for that?

Assessment for the perception of preschool vision screening was measured using the responses of participants in the study. The scale employed was a five-point Likert scale to assess responses of participants. Please see response above under ‘Variable’. We have revised our Statistical Analysis section and provided additional details (please see Revised Manuscript pages 7; lines 130-132).

Assessment for the perception of preschool vision screening was measured using a five-point Likert scale, which elicited needed responses from participants in the study. 

• The authors stated that “Chi-square test was used to determine the association between demographic characteristics of respondents, and their awareness and perception, and the availability of preschool vision screening.” – But no association data is provided for perception, and the availability of preschool vision screening in the results. Moreover, the authors did not run any advance level analysis (bivariate of multi-variate analysis) to establish any solid association between outcome variables and explanatory variables.

We have revised our Statistical Analysis section and included a statement in the Revised Manuscript for clarity. Please see below and Revised Manuscript pages 7; Lines 132 to 133

“Logistic regression analysis further investigated the association between sociodemographic characteristics of respondents and availability, awareness and perception of preschool vision screening.”

• Ethical issues are now mixed-up with the method section. The authors might consider putting the ethical approval and other ethical issues as a sub-heading at the end of the method section.

We thank Reviewer #1 for his/her suggestion under the Methods section. We have now restructured and provided an Ethical Approval Section at the end of the methods (please see below and Revised Manuscript pages 9; lines 159-168).

“Ethical Approval

The study was conducted with adherence to the Declaration of Helsinki, and approval was sought from the Committee on Human Research, Publication & Ethics (CHRPE), Kwame Nkrumah University of Science and Technology (KNUST), Kumasi, Ghana (CHRPE/AP/222/18). Permission was also obtained from the Metropolitan Education Directorate and selected schools in the Kumasi Metropolis, Ghana. A verbal informed consent was obtained from all respondents after the details of the nature of the study were explained to them”. 

Result:

• The findings have not really been fully teased and pondered.

All salient findings have been reported clearly under the Results section (Please see Revised Manuscript pages 9 to 17).

• Whilst I appreciate there is a huge amount of data and much to explore one glaring point of interest to me was further elaboration of results.

We thank Reviewer #1 for his/her suggestion and have provide additional details under the Results Section of our Revised Manuscript (please see Pages 9 to 17).

• The result section only highlighted the basic findings on demographic characteristic and few information on awareness and perception, and the availability of preschool vision screening. No associated factors are provided for perception, and the availability of preschool vision screening.

We have now clarified this in the Results Section (please see pages 12 to 17). 

• The authors did not run any bivariate of multi-variate analysis to establish any strong association between outcome variables and explanatory variables. It is difficult to make any inference based on the results provided at present.

We have provided additional analyses and presented additional results based on our study data (please see Revised Manuscript pages 12 to 17).

“In the logistic regression, private school ownership (Odds Ratio [2.46]; 95% CI [1.34-4.51]) was significantly associated with the availability of preschool vision screening. Thus, respondents from private schools were more likely to report the availability of preschool vision screening in their schools.

The logistic regression analysis showed that age (1.03; 1.01-1.05), private school ownership (1.98; 1.14-3.43), teachers (0.46; 0.22-0.97) and preschool experience of more than 10 years (3.45; 1.35-8.85) were statistically significantly related with the awareness of preschool vision screening. Thus, respondents from private schools, teachers, age, and respondents having more than 10 years of preschool experience had an increased likelihood of being aware of preschool vision screening.

In the logistic regression analysis, there was no statistically significant association between sociodemographic characteristics of respondents and their perception of preschool vision screening.”

• Is it possible to expand the result section further establishing association between outcome variables and explanatory variables through running bivariate of multi-variate analysis?

We thank Reviewer #1 for his/her suggestion under the Results section and have provided additional details accordingly (please see Revised Manuscript pages 12 to 17)

“In the logistic regression, private school ownership (Odds Ratio [2.46]; 95% CI [1.34-4.51]) was significantly associated with the availability of preschool vision screening. Thus, respondents from private schools were more likely to report the availability of preschool vision screening in their schools.

The logistic regression analysis showed that age (1.03; 1.01-1.05), private school ownership (1.98; 1.14-3.43), teachers (0.46; 0.22-0.97) and preschool experience of more than 10 years (3.45; 1.35-8.85) were statistically significantly related with the awareness of preschool vision screening. Thus, respondents from private schools, teachers, age, and respondents having more than 10 years of preschool experience had an increased likelihood of being aware of preschool vision screening.

In the logistic regression analysis, there was no statistically significant association between sociodemographic characteristics of respondents and their perception of preschool vision screening.”

Discussion:

• The discussion should be more analytical and should reflect on how the findings support, refute, extend the previous literature (generally cited in the introduction).

The literature as used in the Introduction section (references 7, 11-16, 18 in the originally submitted manuscript) were also used under the Discussion section to explain findings in our study. We, however, acknowledge other references which were drawn to fully support our arguments under the Discussion section. We have also reviewed some references under the Discussion section and provided additional details (please see Revised Manuscript pages 16 to 19). 

“This was proven by the statistically significant association between teachers and preschool teaching experience of more than 10 years, and the level of awareness of preschool vision screening”.

• Results of previous studies are simply stated alongside this study with no discussion of why the findings may differ between the studies etc.

We have addressed this issue in our Revised Manuscript. Please see Revised Manuscript Page 16 to 19.

• What is implication of the study results for preschool vision screening program and eye care linked clinical practice?

The study assesses and recommends the need for a national preschool vision screening programmes/policies in all countries to preserve the sight/vision of children. Eye care professionals must play an instrumental role in the early assessment of preschoolers through the national preschool vision screening programmes.

• Moreover, the section would benefit from some tightening through building more precise arguments more in the context of this study findings only.

We thank Reviewer #1 for his/her suggestion under the Results section

• The researchers highlight a few limitations of the study which should be further pondered. The strengths of the study should be highlighted as well. The authors should highlight limitations/strengths with strong arguments.

We have now revised the limitations/strengths section of the Discussion with strong arguments. Please see below and Revised Manuscript Page 21, Lines 302 to 309

“The strength of this study lies in the use of an open-ended questionnaire which elicited more details and/or insight from respondents through an infinite number of likely answers. Limitations in this study include a small sample of stakeholders (mainly teachers and educationists) used in this exploratory analysis. Larger sample size should be considered in future researches to enhance the applicability and generalization of results to other locations and countries. Again, the results may be biased and/or skewed towards a section of stakeholders (teachers and educationists only) associated with preschool vision screening in Ghana, and may not represent the views/position of other stakeholders such as parents and eye care professionals.”

This is an interesting topic and I hope the authors will find these comments useful as they consider how to proceed with the manuscript.

We thank the Reviewer for his/her helpful comments regarding our manuscript.

Reviewer #2: I guess the aim of the manuscript is to inform the Ghana Health authority that you are ready to accept any screening policy in case of implementation. In most screening policies implemented throughout the world rest on primary care. Curious to know the health structure in Ghana Do family practitioners perform or would be willing to do the screenings. Any screening program performed in school would be an addition to the primary care physician screening.

In your discussion please state the primary care system in Ghana.

In Ghana, the Ministry of Health is responsible for the health sector. It formulates health policies and coordinates and regulates all stakeholders in the health sector. Implementation of various health policies is carried out by the public, private, and traditional sectors. In the public sector, the Ghana Health Service, Teaching Hospitals Board, and Quasi Government Hospitals are the implementing agencies of the ministry. The private sector, which is regulated by the Private Hospitals and Maternity Homes Board, also comprise mission-based providers, and private medical practitioners.

Primary health care in Ghana comprise practitioners such as physicians and physician associates, nurses, optometrists, pharmacists, etc. Primary health care is delivered in district, municipal, and rural hospitals; who eventually refer ‘bigger’ problems to the higher healthcare institutions such as the Teaching hospitals. The low number of professional healthcare givers in Ghana’s primary healthcare system poses a challenge in addressing the health needs of the people.

The National Eye Care Unit of Ghana is responsible for the eye health needs of the population in Ghana. It is made up of eyecare cadets such as optometrists, ophthalmologists, ophthalmic nurses, and opticians. Optometrists, ophthalmic nurses and opticians serve as primary eyecare providers in Ghana’s eye health system. Ghana currently do not have a national preschool vision screening policy/programme. Most eye care professionals are contracted by school authorities to conduct vision assessment for preschoolers (especially in private schools) outside their daily primary eye care routine in clinics.

Please find attached a picture illustrating the structure of Ghana’s health system.

---

## [Editor Report · Decision Letter 1]

24 Feb 2020

Assessment of Availability, Awareness and Perception of Stakeholders regarding Preschool Vision Screening in Kumasi, Ghana: An Exploratory Study

PONE-D-19-23753R1

Dear Dr. Akuffo,

We are pleased to inform you that your manuscript has been judged scientifically suitable for publication and will be formally accepted for publication once it complies with all outstanding technical requirements.

With kind regards,

Ahmed Awadein, MD, Ph.D, FRCS

Academic Editor

PLOS ONE
---

## [Editor Report · Acceptance letter]

27 Feb 2020

PONE-D-19-23753R1 

Assessment of Availability, Awareness and Perception of Stakeholders regarding Preschool Vision Screening in Kumasi, Ghana: An Exploratory Study 

Dear Dr. Akuffo:

I am pleased to inform you that your manuscript has been deemed suitable for publication in PLOS ONE. Congratulations! Your manuscript is now with our production department. 

With kind regards,

on behalf of

Dr. Ahmed Awadein 

Academic Editor

PLOS ONE